# It Takes Two: Accurate Gait Recognition in the Wild via Cross-granularity Alignment

## ABSTRACT

Existing studies for gait recognition primarily utilized sequences of either binary silhouette or human parsing to encode the shapes and dynamics of persons during walking. Silhouettes exhibit accurate segmentation quality and robustness to environmental variations, but their low information entropy may result in sub-optimal performance. In contrast, human parsing provides fine-grained part segmentation with higher information entropy, but the segmentation quality may deteriorate due to the complex environments. To discover the advantages of silhouette and parsing and overcome their limitations, this paper proposes a novel cross-granularity alignment gait recognition method, named **XGait**, to unleash the power of gait representations of different granularity. To achieve this goal, the XGait first contains two branches of backbone encoders to map the silhouette sequences and the parsing sequences into two latent spaces, respectively. Moreover, to explore the complementary knowledge across the features of two representations, we design the Global Cross-granularity Module (**GCM**) and the Part Cross-granularity Module (**PCM**) after the two encoders. In particular, the GCM aims to enhance the quality of parsing features by leveraging global features from silhouettes, while the PCM aligns the dynamics of human parts between silhouette and parsing features using the high information entropy in parsing sequences. In addition, to effectively guide the alignment of two representations with different granularity at the part level, an elaborate-designed learnable division mechanism is proposed for the parsing features. Finally, comprehensive experiments on two large-scale gait datasets not only show the superior performance of XGait with the Rank-1 accuracy of 81.0% on Gait3D and 88.3% CCPG but also reflect the robustness of the learned features even under challenging conditions like occlusions and cloth changes.

## KEYWORDS

Gait Recognition, In the Wild, Gait Representation, Silhouette, Human Parsing

## 1 INTRODUCTION

Gait, as a distinctive biometric feature, can be used to identify individuals based on their walking patterns. Unlike face and fingerprint, gait has lots of advantages, such as difficulty to disguise, remote

*ACM MM, 2024, Melbourne, Australia*
© 2024 Copyright held by the owner/author(s). Publication rights licensed to ACM.
ACM ISBN 978-x-xxxx-xxxx-x/YY/MM
https://doi.org/10.1145/nnnnnnn.nnnnnnn

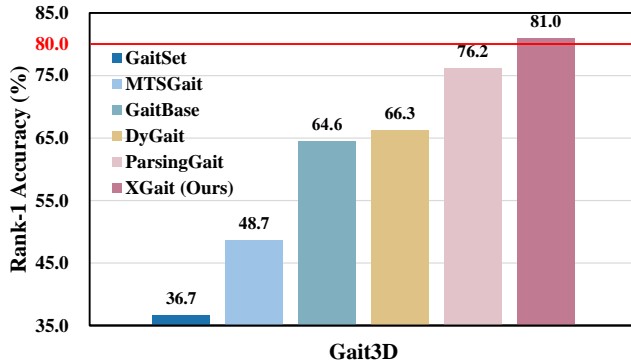

**Figure 1: Comparisons of different gait recognition methods, i.e., GaitSet [4], MTSGait [50], GaitBase [7], DyGait [39], and ParsingGait [52] on the Gait3D dataset in terms of Rank-1 accuracy. (Best viewed in color.)**

accessibility, and non-contact nature. Consequently, gait recognition holds significant utility in intelligent monitoring and social security [24, 34].

In recent years, the studies for gait recognition have begun to transfer from laboratory environments to real-world scenarios [51, 53]. Due to the complex background, occlusion, and cloth-changing in real-world scenarios, the traditional gait recognition methods often perform poorly [16, 51]. According to the recent research progress, we find that the appearance representations, i.e., silhouette and parsing sequences, gradually show certain advantages in this challenging task. For example, HSTL [37] utilizes the hierarchical clustering analysis to obtain multi-scale spatial-temporal gait features from silhouette sequences, which achieves Rank-1 accuracy over 60% on the Gait3D dataset. DyGait [39] proposes a dynamic augmentation module to extract dynamic features of gait from silhouette sequences. ParsingGait [52] for the first time uses parsing sequences as the gait representation and extracts dynamic gait features from human body parts, which obtains a 17.5% Rank-1 increase compared with the State-Of-The-Art (SOTA) gait recognition methods. However, although researchers have made great efforts in these two promising appearance representations, i.e., silhouette and parsing, their performance is still far from practical application.

Silhouette sequences have been dominating the gait recognition community for a long time. Because they are easy to extract from the video frames, and compared to the sparse skeleton sequences, they also contain more useful gait information like appearance and belongings. However, the information entropy of the binary silhouette is still very low, resulting in unsatisfactory performance. Most recently, an emerging gait representation, i.e., gait parsing sequence, is proposed [52]. It can encode the shapes and dynamics

of fine-grained human parts during walking, which can provide higher information entropy. Therefore, more and more researchers gradually pay attention to using parsing sequences for gait recognition. However, extracting parsing requires a more powerful human parsing model. In fact, the complex background, occlusion, and cloth-changing in real-world scenarios make it extremely difficult to train such a robust human parsing model. Consequently, the quality of the parsing is generally lower than that of the silhouette, which hinders it from achieving optimal performance.

To this end, we propose a novel cross-granularity alignment gait recognition framework called **XGait**. It is the first framework to integrate the two most promising appearance representations, i.e., silhouette and parsing sequences, to achieve accurate gait recognition in the wild. The XGait framework comprises three primary stages. First, the silhouette sequence and gait parsing sequence are derived from the original RGB frames through segmentation and human parsing models. Next, we input these two appearance representations into separate CNN-based backbones to extract distinctive gait features. At last, we develop the Global Cross-granularity Module (**GCM**) and Part Cross-granularity Module (**PCM**) to achieve the granular alignment across global and part levels. The GCM is designed to enhance the quality of parsing features through the global appearance features extracted by silhouette sequences. The PCM is devised to align the dynamics of human body parts between silhouette and parsing features based on the high information entropy in parsing sequences. Moreover, to effectively handle the occlusion situation that is easy to appears in real-world scenarios, we propose a learnable division mechanism to guide the alignment of two granularity features at the part level. Finally, our XGait can leverage the superior segmentation quality of silhouette along with the high information entropy provided by parsing. Comprehensive experimental results on Gait3D and CCPG datasets demonstrate the effectiveness of our proposed method.

In summary, the contributions of this paper are as follows:

- To our knowledge, this study is the first work to investigate the integration of two most effective appearance representations, i.e., silhouette sequences and parsing sequences. Following the insightful analysis, we notice and explore their distinct advantages to achieve complementarity.
- We propose a novel cross-granularity alignment gait recognition framework called **XGait** to unleash the power of silhouette and parsing. In XGait, the Global Cross-granularity Module (GCM) and Part Cross-granularity Module (PCM) are designed to achieve efficient granular alignment at the global and part levels, respectively.
- We achieve the state-of-the-art performance on two large-scale gait datasets, i.e., Gait3D and CCPG. Our XGait achieves the Rank-1 accuracy of 81.0% on Gait3D and 88.3% on CCPG, respectively. The experiments reflect that our method is robust under challenging conditions like occlusions and cloth changes.

## 2 RELATED WORK

In this section, the gait representations are first surveyed. Next, we introduce the current gait recognition methods.

### 2.1 Gait Representations

Early gait representations primarily consisted of hand-designed models, such as 3D cylinders [2] and Chrono-Gait Image [35]. However, these hand-designed representations lack critical gait information. Subsequently, researchers began employing the frame subtraction method to extract pedestrian silhouettes and utilized them as the gait representations [45]. Building on this foundation, Han *et al.* [11] introduced the Gait Energy Image (GEI), a compressed representation derived from the silhouette sequence for gait feature extraction. However, the frame subtraction method severely constrains the scope of gait representations that can be extracted. Fortunately, due to the advancements in semantic segmentation and pose estimation algorithms [3, 21, 36], an increasing number of gait datasets utilizing silhouette and skeleton data have emerged. Representative datasets can be categorized into the CASIA series [26, 28, 38, 45] and the OU-ISIR series [1, 12, 15, 23, 31, 32, 41]. Generally, due to its sparsity, skeletal information is not as comprehensive as that provided by silhouettes. The silhouette sequence has long been the predominant choice in the field of gait recognition.

However, the aforementioned datasets are all gathered from controlled laboratory environments, posing challenges to the applicability of gait recognition technology. Recently, researchers have started constructing gait datasets from real-world environments, such as GREW [53] and Gait3D [51]. Challenges such as occlusion, 3D viewpoint changes, irregular walking routes, and various walking speeds in real-world environments cause difficulties for silhouette sequences to satisfy the requirements of the gait recognition task. Consequently, existing gait recognition methods often demonstrate poor performance on in-the-wild datasets [51, 53], despite their success on in-the-lab datasets [15, 27, 28].

Most recently, Zheng *et al.* [52] proposed a novel gait representation called Gait Parsing Sequence (GPS), which exhibits significantly higher information entropy compared to the silhouette sequence. Substituting the silhouette sequence with GPS results in notable performance enhancements in contemporary appearance-based gait recognition methods. However, achieving more refined parsing results necessitates more strict requirements for human parsing models. This will result in a reduction in segmentation quality. Conversely, despite the low information entropy of silhouettes, their quality is ensured due to their straightforward characteristics. In general, silhouette and parsing, as promising appearance representations, each has its own strengths and limitations. We believe that integrating them will effectively improve gait recognition performance, which has not been deeply explored in the gait recognition community.

### 2.2 Gait Recognition Methods

In this section, we divide the existing gait recognition methods into single-representation and multi-representation approaches. Single-representation methods occupy the majority of the community and can also be divided into two main categories: model-based methods and appearance-based approaches [34]. In earlier years, model-based approaches dominated the field of gait recognition, including the 2D/3D skeletons [17], a structural human body model [2, 42], etc. For example, Urtasun *et al.* [33] introduced a gait analysis technique that leverages articulated skeletons and

three-dimensional temporal motion models. Zhao *et al.* [49] employed a local optimization algorithm to pursue motion tracking for gait recognition. Yamauchi *et al.* [43] proposed an interesting approach for human walking recognition by utilizing 3D pose estimation derived from RGB frames. Nevertheless, model-based techniques are less potent than appearance-based methods in gait recognition due to their inability to capture relevant gait features such as shape and appearance. Appearance-based approaches, also known as model-free methods, often utilize the gait silhouette sequence, gait parsing sequence and Gait Energy Image (GEI) as input [4, 9, 11, 13, 14, 18, 19, 25, 40, 47, 48, 52]. For example, Han *et al.* suggested the consolidation of a series of silhouettes into a concise Gait Energy Image (GEI) [11] and extracted gait knowledge from it. Chao *et al.* [4] treated the gait sequence as an unordered set and employed Convolutional Neural Networks (CNNs) to extract gait features from the frame-level and sequence-level. Fan *et al.* [9] proposed the GaitPart framework to horizontally split silhouettes and extract fine-grained features from each part. However, despite their impressive performance in controlled laboratory environments, these methods often falter when applied to real-world scenarios [51, 53]. In fact, silhouettes exhibit low information entropy, which renders them incapable of fully capturing the complexity of gait information in real-world scenarios.

Most recently, the Gait Parsing Sequence (GPS) has been introduced as a novel gait representation that not only captures the body shape and appearance but also encompasses the dynamics of fine-grained human body parts [52]. By replacing the input from the silhouette sequence with the GPS, existing appearance-based methods achieve a significant performance improvement, i.e., 12.5% ~ 19.2% improvements in Rank-1 accuracy. Furthermore, Zheng *et al.* [52] proposed the ParsingGait network to extract the discriminative gait features from the global appearance information and the relations among human body parts. However, compared to silhouette, parsing necessitates a higher requirement for semantic segmentation models, i.e., human parsing models. Especially, through experiments on another challenging cloth-changing dataset, i.e., CCPG, we observe that the performance of parsing is inferior to the silhouette due to its low quality. Morte details can be found in Section 5.

In recent years, multi-representation methods have been studied in the gait recognition community. For instance, Cui *et al.* [5] introduced the MMGaitFormer framework to realize the effective spatial-temporal feature fusion from silhouette and skeleton sequences. Fan *et al.* [8] proposed the SkeletonGait++ model to integrate the skeleton map and silhouette features. However, the fusion between silhouette and parsing sequences, the two most promising appearance representations, has not been studied by gait recognition researchers. This paper aims to explore their complementarity to achieve accurate gait recognition in the wild.

## 3 METHOD

This section provides a detailed introduction to the proposed XGait framework. First, we provide an overview of our method in Section 3.1. Next, we discuss the silhouette and parsing encoding modules in Section 3.2. In Section 3.3, we explain how our Global Cross-granularity Module (GCM) utilizes the global appearance features

extracted from silhouette sequences to improve the quality of parsing features. In Section 3.4, we introduce our Part Cross-granularity Module (PCM) and the designed learnable division mechanism, and explain how it establishes the fine-grained alignment between the silhouette and parsing features. Finally, we present the details of training and inference.

### 3.1 Overview

The architecture of the proposed gait recognition framework, i.e., XGait, is illustrated in Figure 2. The framework consists of three main stages: the preprocessing stage, the encoding stage, and the cross-granularity alignment stage.

In the preprocessing stage, two appearance representations will be extracted offline from the original RGB sequence. One is the silhouette sequence $S \in \mathbb{R}^{T \times C \times H \times W}$ obtained using a semantic segmentation model. Here, $T$ represents the sequence length, while $C$, $H$, and $W$ denote the channel, height, and width of the frame, respectively. The other representation is the gait parsing sequence $P \in \mathbb{R}^{T \times C \times H \times W}$, extracted using a human parsing model.

In the encoding stage, we utilize two separate CNN-based backbone networks to extract the distinctive characteristics of each appearance representation. Specifically, we obtain the feature maps $\mathbf{F}_s$ and $\mathbf{F}_p$ from the silhouette sequence $S$ and gait parsing sequence $P$, respectively. Here, $\mathbf{F}_s$ and $\mathbf{F}_p$ belong to $\mathbb{R}^{T \times c \times h \times w}$, with $c$, $h$, and $w$ representing the channel, height, and width of the feature maps.

In the cross-granularity stage, the above two feature maps are fed into two branches: (1) the Global Cross-granularity Module (GCM) integrates these two granularity features at the global level, aiming to optimize the parsing features with relatively low segmentation quality through the global silhouette features. This global-level integration results in the feature map $\widehat{F}_{ga}$. (2) the Part Cross-granularity Module (PCM) aligns the dynamics of human parts between silhouette and parsing features using the high information entropy in parsing, yielding the feature map $\widehat{F}_{pa}$.

Then, by merging $\mathbf{F}_s$ and $\mathbf{F}_p$, we employ four separate Feature Mapping Heads (FMHs) to further extract more discriminative features, resulting in features $\widehat{\mathbf{F}}_s$, $\widehat{\mathbf{F}}_p$, $\widehat{\mathbf{F}}_{ga}$, and $\widehat{\mathbf{F}}_{pa}$. Subsequently, concatenating these features along the channel dimension yields $\widehat{\mathbf{F}}_{out}$, as the final output feature.

Finally, the cross-entropy loss $L_{ce}$ and the triplet loss $L_{tri}$ are employed to train the model.

### 3.2 Silhouette and Parsing Encoding Module

While both the silhouette sequence and parsing sequence are appearance representations, they possess distinct characteristics: 1) Semantic information: Parsing provides richer semantic details compared to the binary silhouette, including delineation of the head, left/right hand, etc. 2) Segmentation quality: The silhouette exhibits more accurate segmentation quality due to its simplicity and more robustness to environmental variations.

Based on the above insights, we utilize two separate encoders to extract distinctive features from these two appearance representations. In particular, the silhouette encoding module $F_S(\cdot)$ and the parsing encoding module $F_P(\cdot)$ are established with ResNet-like structures [7]. The silhouette feature maps $\mathbf{F}_s$ and the parsing feature maps f$\mathbf{F}_p$ are then obtained from silhouette sequences and

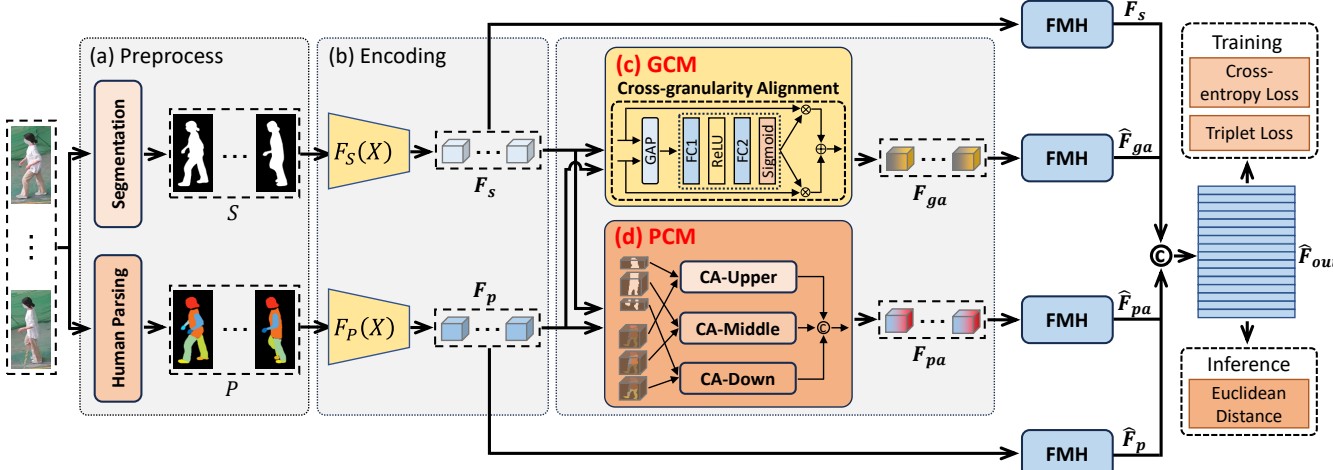

**Figure 2: The architecture of our XGait. In the preprocessing stage, the silhouette sequence $S \in \mathbb{R}^{T \times C \times H \times W}$ and the parsing sequence $P \in \mathbb{R}^{T \times C \times H \times W}$ are extracted from the RGB sequences by segmentation method and human parsing model, respectively. In the Encoding stage, we employ two separate ResNet-like structure backbones $F_S(\cdot)$ and $F_P(\cdot)$ to extract the mid-level features from the silhouette sequence and the parsing sequence, respectively. In the Cross-granularity stage, the Global Cross-granularity Module (GCM) and the Part Cross-granularity Module (PCM) are proposed to explore the complementary knowledge from these two granularity features across global and part levels. GAP denotes the Global Average Pooling. FC means the Fully Connected layer. CA refers to the Cross-granularity Alignment module. FMH represents the Feature Mapping Head.**

parsing sequences, respectively. it is worth noting that the backbone within the Silhouette and Parsing Encoding Modules can be substituted with any other gait recognition network. Although a stronger backbone may yield performance enhancements, this is not the focus of our method.

### 3.3 Global Cross-granularity Module

The Global Cross-granularity Module (GCM) aims to utilize the global appearance features of the silhouette to improve the quality of parsing features. As illustrated in Figure 2 (c), a light-weighted module, i.e., Cross-granularity Alignment (CA) module, is designed to align $\mathbf{F}_s$ and $\mathbf{F}_p$ at the global level. To preserve the distinct semantic information, we concatenate $\mathbf{F}_s$ and $\mathbf{F}_p$ along the channel dimension. Before that, we employ Global Average Pooling (GAP) to compress the spatial dimension for computational consideration. The process is formulated as follows:

$$\mathbf{F}_{sp} = [GAP(\mathbf{F}_s), GAP(\mathbf{F}_p)], \tag{1}$$

where $[\cdot, \cdot]$ represents the concatenation operation.

Subsequently, we apply two Fully Connected (FC) layers, each with $c/r$ and $2c$ neurons, respectively, where $r$ denotes the reduction ratio. Following the first FC layer, we employ the ReLU activation function. Then, a sigmoid layer is utilized after the final FC layer to derive element-wise weights for the silhouette feature $F_s$ and the parsing feature $F_p$. Finally, the output feature $\mathbf{F}_{ga}$ is obtained through element-wise weighted summation. These operations facilitate interaction between silhouette and parsing features at a global level. The described operations can be expressed as:

$$\mathbf{F}_{ga} = f_{ca}^1(\mathbf{F}_{sp}) \cdot \mathbf{F}_s + f_{ca}^2(\mathbf{F}_{sp}) \cdot \mathbf{F}_p, \tag{2}$$

where $f_{ca}^1(\cdot)$ and $f_{ca}^2(\cdot)$ represent the outputs of $f_{ca}(\cdot)$ splitting along the channel dimension, with $f_{ca}^1(\cdot)$ containing the first half and $f_{ca}^2(\cdot)$ containing the second half. In addition, the detailed operation of the CA module can be expressed as follows:

$$f_{ca}(\mathbf{F}_{sp}) = Sigmoid(\mathbf{W}_2 \sigma(\mathbf{W}_1 \mathbf{F}_{sp}))$$
$$= [f_{ca}^1(\mathbf{F}_{sp}), f_{ca}^2(\mathbf{F}_{sp})], \tag{3}$$

where $\mathbf{W}_1$ and $\mathbf{W}_2$ are the weights of the two FC layers, $\sigma$ denotes the ReLU activation function.

### 3.4 Part Cross-granularity Module

The Part Cross-granularity Module (PCM) is designed to take advantage of the high information entropy in parsing to align human body parts between silhouette and parsing features, as is shown in Figure 2 (d). In general, the human body can generally be divided into three main parts: 1) the upper body, which includes the head and shoulders; 2) the middle body, comprising the torso, arms, and hands; 3) the lower body, encompassing the legs and feet. Meanwhile, previous studies [5, 9] have shown that different parts of the human body display distinct movement patterns during walking.

Therefore, we constrain the silhouette and parsing features to capture mutual information associated with the relevant body parts. As illustrated in Figure 4, we establish a part-level relationship between the silhouette and parsing features. The silhouette feature $\mathbf{F}_s$ is horizontally divided into three segments: the top quarter $(0 - \frac{1}{4})$, the middle half $(\frac{1}{4} - \frac{3}{4})$, and the bottom quarter $(\frac{3}{4} - 1)$. Similarly, following the categories of human body parts in [52], we segment the parsing feature $\mathbf{F}_p$ into three regions: the upper body (head), the middle body (torso, left arm, right arm, left hand, right hand, dress), and the lower body (left leg, right leg, left foot, right foot,

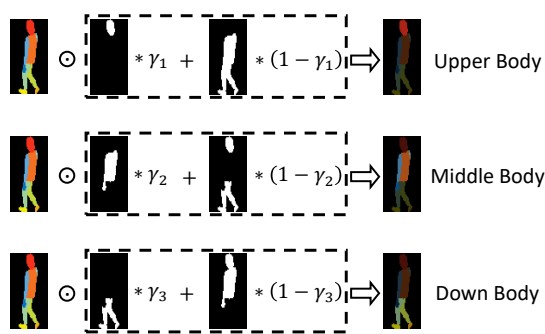

Figure 3: The illustration of the learnable division. $\gamma$ is the learnable parameter used to modulate the weight of the i-th body part. (Best viewed in color.)

dress). Acknowledging that certain human body parts, such as the head, hands, and feet, are prone to occlusion due to their small size, we propose a learnable division mechanism to segment the parsing regional features, as depicted in Figure 3. The mathematical expression for the learnable division is as follows:

$$\mathbf{F}_p^i = \gamma_i * \mathbf{F}_p \odot M_i + (1 - \gamma_i) * \mathbf{F}_p \odot (1 - M_i), \quad (4)$$

where $i = 1, 2, 3$ is the index of the upper body, the middle body, and the lower body, respectively. $M_i$ denotes the mask corresponding to the i-th human body part. $\gamma_i$ represents a learnable parameter used to modulate the weight of the feature associated with the i-th body part, and $\odot$ denotes the element-wise multiplication.

As is shown in Figure 4, after obtaining distinct part-level silhouette and parsing features, we apply average pooling and max pooling to reduce the spatial dimension. Next, three independent CA modules, namely CA-Upper, CA-Middle, and CA-Down, are employed to extract the Cross-granularity features $\mathbf{F}_{pa}^i$. Lastly, we concatenate the $\mathbf{F}_{pa}^i$ horizontally to obtain the output feature $\mathbf{F}_{pa}$ of PCM. In this way, our XGait can fully leverage the complementary advantages of these two appearance representations, resulting in more robust and discriminative features for gait recognition.

### 3.5 Feature Mapping Head

Four Feature Mapping Heads (FMHs) are implemented to enhance the efficiency of training and inference. These heads, applied to the discriminative gait features $\mathbf{F}_s$, $\mathbf{F}_p$, $\mathbf{F}_{ga}$, and $\mathbf{F}_{pa}$, utilize Set Pooling (SP) and Horizontal Pyramid Mapping (HPM) techniques [4]. The SP compresses temporal knowledge using max pooling, significantly reducing computational complexity. The HPM consists of Horizontal Pyramid Pooling (HPP) and Separate Mapping (SM). The HPP performs fine-grained sampling and horizontally reduces the dimension using max pooling and mean pooling. The SM utilizes separate fully connected layers for each pooled feature to aggregate more discriminative information. Taking $\mathbf{F}_s$ as an example, the above operations can be expressed as:

$$\widehat{\mathbf{F}}_s = H(P(\mathbf{F}_s)), \quad (5)$$

Where $P(\cdot)$ represents the SP operation, $H(\cdot)$ denotes the HPM process.

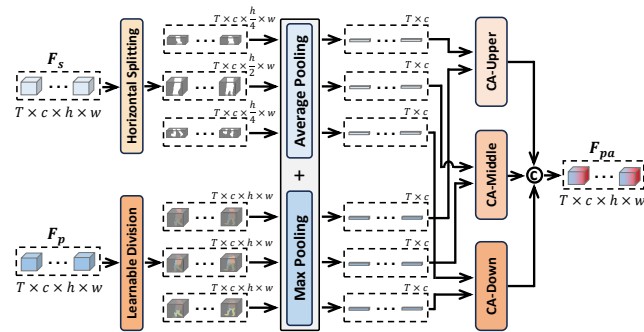

Figure 4: The pipeline of the Part Cross-granularity Module (PCM). CA-Upper, CA-Middle, and CA-Down are three independent Cross-granularity Alignment modules for extracting fine-grained mutual information from various human body parts. (Best viewed in color.)

At last, we concatenate $\widehat{\mathbf{F}}_s$, $\widehat{\mathbf{F}}_p$, $\widehat{\mathbf{F}}_{ga}$, and $\widehat{\mathbf{F}}_{pa}$ to obtain the final feature vector $\widehat{\mathbf{F}}_{out}$, which is used for sequence-to-sequence matching during both training and inference. For a comprehensive understanding of the SP and HPM, please refer to [4].

### 3.6 Training and inference

Our XGait framework is trained in an end-to-end manner. We optimize the network through a loss function comprising two components:

$$L = \alpha L_{tri} + \beta L_{ce}, \quad (6)$$

where $L_{tri}$ is the triplet loss, $L_{ce}$ is the cross entropy loss. $\alpha$ and $\beta$ are the weighting parameters.

In the inference phase, the similarity between a query-gallery pair is assessed by computing the Euclidean distance.

## 4 EXPERIMENTS

We first introduce the datasets used in this work in Section 4.1. Furthermore, implementation details are outlined in Section 4.2. Section 4.3 presents a comprehensive comparison of our proposed XGait with existing gait recognition methods. Then, ablation studies on our framework are conducted in Section 4.4. At last, the visualization of heatmaps is provided in Section 4.5.

### 4.1 Datasets

**Gait3D.** The Gait3D dataset, proposed in [51], is a challenging gait recognition dataset derived from real-world scenarios. It comprises 4,000 subjects, 25,309 sequences, and 3,279,239 frame images captured by 39 cameras in an unconstrained indoor environment, i.e., a large supermarket. To comply with the official train/test strategy outlined in [51], 3,000 subjects are chosen for training, while another 1,000 subjects are reserved for testing. Moreover, each subject in the testing set has one sequence designated as the query, while the remaining sequences are used as the gallery. The Rank-1, Rank-5, mean Average Precision (mAP), and mean Inverse Negative Penalty (mINP) [44] are employed to evaluate the performance of the Gait3D dataset.

**CCPG.** The CCPG dataset [16] is a recently introduced dataset designed for cloth-changing gait recognition. It consists of 200 subjects and 16,566 sequences captured using two outdoor cameras and eight indoor cameras. It provides a diverse array of outfits, encompassing 13 tops, 8 bottoms, and 5 distinct bags. Moreover, it includes a variety of walking routes, such as turning, occlusion, and background change routes. The above factors also make it an extremely challenging gait dataset. There are 100 subjects for training and another 100 for testing. In the testing phase, three types of cloth-changing scenarios are provided: cloth-changing (CL-FULL), ups-changing (CL-UP), and pants-changing (CL-DN). Rank-1 accuracy and mean Average Precision (mAP) are adopted as evaluation metrics to assess the performance of the CCPG dataset.

## 4.2 Implementation Details

**Input.** We utilize the officially provided silhouette and parsing data directly as inputs to our XGait model for the Gait3D dataset. For the CCPG dataset, where parsing is unavailable, we utilize CDGNet [20] to extract the parsing information. To ensure basic quality standards, we randomly sample and label 1,400 RGB images from the CCPG dataset. Labeling follows the guidelines outlined in [52]. The CDGNet model is then fine-tuned using the parameters published in [52]. Finally, we utilize the optimized CDGNet model to extract the parsing data from the CCPG dataset.

**Setting.** Following the same preprocessing in [4], silhouette and parsing images are normalized to $64 \times 44$ for both Gait3D and CCPG datasets. The silhouette and parsing encoding module employ the GaitBase [7] as the backbone. For the Gait3D dataset, the batch size is $32 \times 2 \times 30$, representing 32 subjects, 2 sequences per subject, and 30 frames per sequence. The reduction ratio $r$ in the GCM is 1. The batch size for the CCPG dataset is $8 \times 8 \times 30$. The reduction ratio $r$ is set to 16. Training the model for both Gait3D and CCPG datasets involves 1,200K iterations. The Learning Rate (LR) starts at 0.1 and is subsequently multiplied by 0.1 at the 400K, 800K, and 1,000K iterations. The SGD optimizer is adopted with a weight decay of 5e-4. The unordered sampling strategy is implemented in both datasets. The margin in the triplet loss is 0.2. In Equation 6, the weighting parameters $\alpha$ and $\beta$ are set to 1.0. During the inference phase, all frames of each gait sequence are utilized, with a maximum of 720 frames per sequence for memory consideration.

## 4.3 Comparison with State-Of-The-Art Methods

In this section, we compare the proposed XGait with several popular state-of-the-art (SOTA) gait recognition methods.

**Evaluation on Gait3D.** The experimental results on the Gait3D dataset are listed in Table 1. It is evident that model-based approaches, such as PoseGait and GaitGraph, typically underperform in comparison to appearance-based methods. This is because the utilization of sparse keypoints of the human body, leads to a deficiency in comprehensive gait information, further exacerbated by the complexity of real-world scenarios. In contrast, appearance-based methods tend to exhibit significantly superior performance. However, GEINet's performance is also poor due to significant information loss during the compression of the gait silhouette sequence into a gait energy image. Moreover, the approach utilizing parsing

**Table 1: Comparison of the SOTA gait recognition methods on the Gait3D dataset. R-1 and R-5 denote the Rank-1 and Rank-5 accuracy, respectively.**

| Methods | Publication | R-1 | R-5 | mAP | mINP |
|---|---|---|---|---|---|
| PoseGait [17] | PR 2020 | 0.2 | 1.1 | 0.5 | 0.3 |
| GaitGraph [30] | ICIP 2021 | 6.3 | 16.2 | 5.2 | 2.4 |
| GPGait [10] | ICCV 2023 | 22.5 | - | - | - |
| GEINet [25] | ICB 2016 | 5.4 | 14.2 | 5.1 | 3.1 |
| GaitSet [4] | AAAI 2019 | 36.7 | 58.3 | 30.0 | 17.3 |
| GaitPart [9] | CVPR 2020 | 28.2 | 47.6 | 21.6 | 12.4 |
| GLN [13] | ECCV 2020 | 31.4 | 52.9 | 24.7 | 13.6 |
| GaitGL [19] | ICCV 2021 | 29.7 | 48.5 | 22.3 | 13.3 |
| CSTL [14] | ICCV 2021 | 11.7 | 19.2 | 5.6 | 2.6 |
| SMPLGait [51] | CVPR 2022 | 46.3 | 64.5 | 37.2 | 22.2 |
| MTSGait [50] | ACM MM 2022 | 48.7 | 67.1 | 37.6 | 21.9 |
| DANet [22] | CVPR 2023 | 48.0 | 69.7 | - | - |
| GaitGCI [6] | CVPR 2023 | 50.3 | 68.5 | 39.5 | 34.3 |
| GaitBase [7] | CVPR 2023 | 64.6 | - | - | - |
| HSTL [37] | ICCV 2023 | 61.3 | 76.3 | 55.5 | 34.8 |
| DyGait [39] | ICCV 2023 | 66.3 | 80.8 | 56.4 | 37.3 |
| ParsingGait [52] | ACM MM 2023 | 76.2 | 89.1 | 68.2 | 41.3 |
| XGait | Ours | **81.0** | **91.9** | **73.3** | **55.4** |

**Table 2: Comparison of the SOTA gait recognition methods on the CCPG dataset. CL-FULL, CL-UP, and CL-DN denote cloth-changing, ups-changing, and pants-changing, respectively.**

| Method | Publication | CL-FULL | | CL-UP | | CL-DN | |
|---|---|---|---|---|---|---|---|
| | | R-1 | mAP | R-1 | mAP | R-1 | mAP |
| GaitGraph2 [29] | CVPRW 2022 | 5.0 | 2.4 | 5.7 | 4.0 | 7.3 | 4.2 |
| GaitTR [46] | ES 2023 | 24.3 | 9.7 | 28.7 | 16.1 | 31.1 | 16.4 |
| SkeletonGait [8] | AAAI 2024 | 52.4 | 20.8 | 65.4 | 35.8 | 72.8 | 40.3 |
| GaitSet [4] | AAAI 2019 | 77.7 | 46.4 | 83.5 | 59.6 | 83.2 | 61.4 |
| GaitPart [9] | CVPR 2020 | 77.8 | 45.5 | 84.5 | 63.1 | 83.3 | 60.1 |
| GaitGL [19] | ICCV 2021 | 69.1 | 27.0 | 75.0 | 37.1 | 77.6 | 37.6 |
| OGBase [7] | CVPR 2023 | 78.4 | 44.5 | 82.3 | 58.3 | 86.0 | 59.3 |
| AUG-OGBase [16] | CVPR 2023 | 84.7 | 52.9 | 88.4 | 67.5 | 89.4 | 67.9 |
| XGait | Ours | **88.3** | **59.5** | **91.8** | **74.3** | **92.9** | **75.7** |

as input, namely ParsingGait, outperforms methods employing silhouettes such as GaitSet, GaitBase, and DyGait. This demonstrates the value of the high information entropy contained in the gait parsing sequence, offering more useful gait knowledge. Finally, our XGait achieves optimal performance, reaching a Rank-1 accuracy of 81%, showcasing the effectiveness of combining silhouette and parsing sequences harmoniously. Meanwhile, this illustrates the significant potential of our method for practical application in real-world scenarios.

**Evaluation on CCPG.** Table 2 presents the experimental results obtained from the CCPG dataset, also known as the cloth-changing gait dataset. We can first observe that model-based approaches, such as GaitGraph2 [29], GaitTR [46], and SkeletonGait [8], exhibit comparatively low performance. This once more demonstrates that sparse skeletons make it difficult for the model to acquire enough gait knowledge due to information loss. In contrast, appearance-based methods achieve better results, while our XGait method exhibits superior performance, confirming its effectiveness in tackling challenging scenarios like cloth-changing.

**Table 3: Analysis of Silhouette (Sil.) and Parsing (Par.) on the Gait3D dataset.**

| Methods | Rank-1 | Rank-5 | mAP | mINP |
|---|---|---|---|---|
| Only Sil. | 58.7 | 76.2 | 49.5 | 27.9 |
| Only Par. | 71.2 | 87.3 | 64.1 | 38.1 |
| Ours | **81.0** | **91.9** | **73.3** | **55.4** |

**Table 4: Analysis of different fusion modes on the Gait3D dataset.**

| Methods | Rank-1 | Rank-5 | mAP | mINP |
|---|---|---|---|---|
| Distance Fusion | 75.6 | 89.3 | 68.2 | 50.2 |
| Feature Fusion | 77.0 | 89.8 | 69.7 | 51.2 |
| Ours | **81.0** | **91.9** | **73.3** | **55.4** |

## 4.4 Ablation Study

This section presents ablation studies on key components. First, we analyze the impact of silhouette and parsing on gait recognition. Next, we examine the influence of different fusion modes. Subsequently, we demonstrate the significant performance enhancement achieved by the proposed GCM and PCM modules. Then, we investigate the shareability of the backbone and the mapping head. At last, different part division strategies are evaluated.

**Analysis of Silhouette and Parsing.** As shown in Table 3, using only silhouette as input results in a Rank-1 accuracy of 58.7%. However, replacing silhouette with parsing leads to a notable performance improvement, with Rank-1 accuracy increasing by 12.5%. This improvement can be attributed to parsing sequences having higher information entropy, offering more useful knowledge for gait recognition. In addition, our approach effectively integrates these two appearance representations, resulting in a significant performance enhancement. Specifically, compared to only using silhouette and parsing, the Rank-1 accuracy improved by 22.3% and 9.8%, respectively. For a detailed discussion on silhouette and parsing, please refer to Section 5.

**Fusion mode.** We also assess the impact of various fusion modes, as listed in Table 4. We first implement the distance fusion experiment, where the distance matrices of silhouette and parsing are directly combined and subsequently utilized for metric evaluation. This fusion technique is commonly employed in challenge competitions. When combined with Table 3, it becomes evident that the distance fusion method also leads to performance improvements. In addition, we perform a feature fusion experiment by directly concatenating the two types of gait features after the backbone in the channel dimension. In other words, this is equivalent to removing the GCM and PCM modules from our XGait method. Surprisingly, we observe a significant performance improvement even with such a simple concatenation operation, resulting in Rank-1 and mAP increasing to 77.0% and 69.7%, respectively. This suggests a strong complementarity between silhouette and parsing representations. Furthermore, the fusion mode presented in our method enhances the exploration of mutual information between them, resulting in further improvements in Rank-1 and mAP accuracy, reaching 81.0% and 73.3%, respectively.

**Table 5: Analysis of the GCM and PCM on the Gait3D dataset.**

| Baseline | GCM | PCM | Rank-1 | Rank-5 | mAP | mINP |
|---|---|---|---|---|---|---|
| ✓ | | | 77.0 | 89.8 | 69.7 | 51.2 |
| ✓ | ✓ | | 79.0 | 90.8 | 71.9 | 53.6 |
| ✓ | | ✓ | 80.4 | 90.5 | 71.8 | 53.6 |
| ✓ | ✓ | ✓ | **81.0** | **91.9** | **73.3** | **55.4** |

**Table 6: Analysis of the shareability of the backbone and the mapping head on Gait3D. FMH denotes the feature mapping head. Sha. means that the parameters are shared, and Ind. represents that the parameters are independent.**

| Backbone | FMH | Rank-1 | Rank-5 | mAP | mINP |
|---|---|---|---|---|---|
| Sha. | Sha. | 40.3 | 59.4 | 31.0 | 18.4 |
| Sha. | Ind. | 39.4 | 58.1 | 29.3 | 17.0 |
| Ind. | Sha. | 78.9 | 90.7 | 72.0 | 53.7 |
| Ind. | Ind. | **81.0** | **91.9** | **73.3** | **55.4** |

**Impact of GCM and PCM.** Subsequently, we analyze the impact of the GCM and PCM modules in our XGait framework. The results are presented in Table 5. Both GCM and PCM effectively enhance various evaluation metrics. Additionally, incorporating both modules results in further performance improvement. This highlights the validity of our proposed feature interaction approach from both global and part levels.

**Analysis of the shareability of backbone and mapping head.** We also investigate the shareability of the backbone and the Feature Mapping Head (FMH). As is shown in Table 6, we observe that performance is worst when backbone parameters are shared while FMH parameters are independent. This is because the differing semantics of silhouette and parsing. Sharing backbone parameters can interfere with model learning. Furthermore, independent FMH parameters exacerbate subsequent feature learning. When backbone parameters are independent, even if subsequent FMH parameters are shared, the model's performance is almost doubled. Finally, when both the backbone and FMH parameters are independent, the model achieves its best performance, with a Rank-1 accuracy of 81.0%.

**Analysis of the division strategy.** Additionally, we perform experiments to validate the effectiveness of the proposed learnable division mechanism described in Section 3.4. Keeping other settings constant, we explore three distinct part division strategies: simple division, fixed division, and learnable division. We utilize Equation 4 and Figure 3 to depict the distinctions among the three division strategies. Simple division and fixed division correspond to setting the learnable parameters $\gamma$ to 1 and 0.75, respectively. The best results, as shown in Table 7, are obtained when employing learnable division in the PCM. We also examine the specific values of the three learnable parameters in Equation 4 once the model converges. The final values of $\gamma_1$, $\gamma_2$, and $\gamma_3$ are 1.1, 2.0, and 1.4, respectively. This indicates that the model automatically intensifies the search in the target area. Comparatively, the upper body receives the least attention, while the middle body and lower body receive more attention.

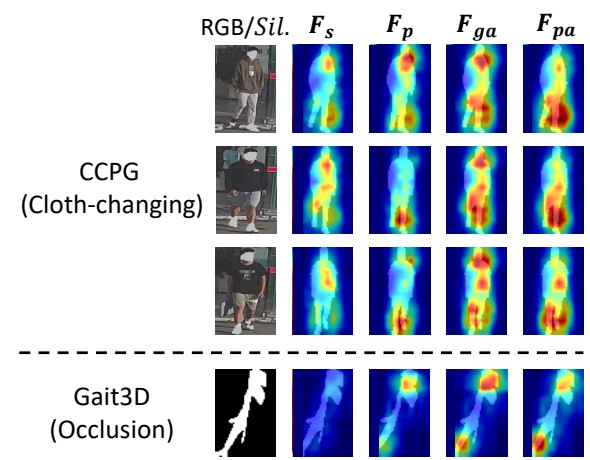

RGB/*Sil.* $\mathbf{F}_s$ $\mathbf{F}_p$ $\mathbf{F}_{ga}$ $\mathbf{F}_{pa}$

CCPG
(Cloth-changing)

Gait3D
(Occlusion)

Figure 5: The heatmaps of $\mathbf{F}_s$, $\mathbf{F}_p$, $\mathbf{F}_{ga}$, and $\mathbf{F}_{pa}$ in our XGait framework. *Sil.* denotes Silhouette. (Best viewed in color.)

Table 7: Analysis of different division strategies on Gait3D.

| Methods | Rank-1 | Rank-5 | mAP | mINP |
|---|---|---|---|---|
| Simple division | 79.5 | 90.9 | 72.6 | 54.9 |
| Fixed division | 79.3 | 91.7 | 71.9 | 54.2 |
| Learnable division | **81.0** | **91.9** | **73.3** | **55.4** |

## 4.5 Visualization

In this sectoin, we visualize the heatmaps of $\mathbf{F}_s$, $\mathbf{F}_p$, $\mathbf{F}_{ga}$, and $\mathbf{F}_{pa}$ in the cloth-changing and occlusion scenarios. As shown in Figure 5, the feature map $\mathbf{F}_{ga}$ focuses on the global region, indicating the GCM module's capability to enhance granularity features at the global level. The $\mathbf{F}_{pa}$ concentrates on the lower body, which indicates that the movement of the legs and feet can effectively reflect useful gait information.

## 5 DISCUSSION

In this section, we aim to thoroughly analyze the influence of silhouette and parsing on gait recognition. As indicated in Table 8, we are surprised to find that parsing shows a lower Rank-1 accuracy compared to silhouette on the CCPG dataset, with 81.4% accuracy compared to 83.9% accuracy with silhouette. Our analysts attribute this to the lower segmentation quality of parsing compared to the silhouette on the CCPG dataset. To verify this hypothesis, we conduct an operation where we intersect the silhouette and parsing, as depicted in Figure 6. This operation ensures that the silhouette and parsing edges are identical, which effectively controls the segmentation quality variable.

Subsequently, we conduct experiments utilizing the intersecting silhouette ($Sil.^*$) and the intersecting parsing ($Par.^*$), finding that parsing yields superior results compared to the silhouette. Specifically, after the contour edges are aligned, the Rank-1 with the parsing is 4.1% higher than with the silhouette. This demonstrates that parsing's high information entropy significantly benefits gait recognition, even in cloth-changing scenarios.

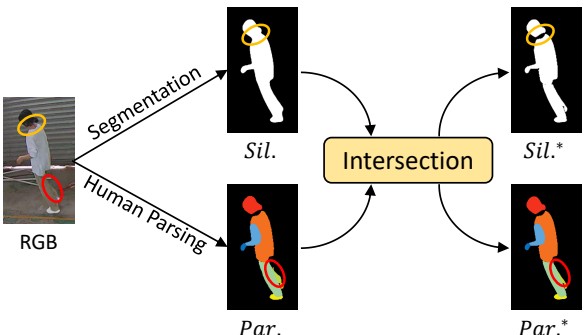

Figure 6: The illustration of the silhouette and parsing intersection operations. *Sil.* denotes Silhouette, *Par.* represents Parsing, $Sil.^*$ signifies the silhouette result after the intersection of *Sil.* and *Par.*, $Par.^*$ denotes the parsing result after the intersection of *Sil.* and *Par.*. (Best viewed in color.)

Table 8: Detailed analysis of silhouette and parsing results on the CCPG dataset. CL-FULL, CL-UP, and CL-DN denote cloth-changing, ups-changing, and pants-changing, respectively.

| Method | CL-FULL | | CL-UP | | CL-DN | |
|---|---|---|---|---|---|---|
| | R-1 | mAP | R-1 | mAP | R-1 | mAP |
| GaitBase (*Sil.*) | 83.9 | 53.4 | 89.6 | 69.4 | 90.1 | 70.9 |
| GaitBase (*Par.*) | 81.4 | 52.3 | 87.3 | 67.7 | **90.8** | **71.3** |
| GaitBase ($Sil.^*$) | 78.3 | 49.6 | 86.2 | 65.8 | 88.2 | 68.5 |
| GaitBase ($Par.^*$) | 82.4 | 53.2 | 87.7 | 68.2 | 90.8 | 72.0 |
| XGait ($Sil.^*$, $Par.^*$) | 82.5 | 53.3 | 87.6 | 69.0 | 90.1 | 71.4 |
| XGait (*Sil.*, *Par.*) | **88.3** | **59.5** | **91.8** | **74.3** | **92.9** | **75.7** |

Finally, we replace the (*Sil.*, *Par.*) with ($Sil.^*$, $Par.^*$) as inputs to our XGait framework. The experimental results indicate a notable performance reduction. In particular, the Rank-1 accuracy decreased from 88.3% to 82.5%. This is because forcing segmentation edge alignment results in the loss of the advantage of segmentation quality in silhouette sequences. In contrast, the XGait with original silhouette and parsing sequences effectively leverages the complementary information of these two appearance representations, realizing the best gait recognition performance.

## 6 CONCLUSION

In this paper, we propose a novel cross-granularity gait recognition framework, named XGait, to unleash the power of different appearance representations, i.e., silhouette and parsing sequences. The Global Cross-granularity Module (GCM) and Part Cross-granularity Module (PCM) are developed to explore the complementary knowledge across the features of these two distinctive representations. Extensive experiments on two large-scale datasets, Gait3D and CCPG, validate the effectiveness of our method under challenging conditions like occlusions and cloth changes. In the future, further exploration of information interaction across more gait representations, such as silhouettes, parsing maps, 3D meshes, skeletons, etc., is expected to promote the development of gait recognition.

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
