# OpenReview forum: "It Takes Two: Accurate Gait Recognition in the Wild via Cross-granularity Alignment"
_acmmm.org/ACMMM/2024/Conference — MM2024 Poster_

### Official Review · Reviewer_cJ4m · 2024-05-22

**Rating:** 5
**Confidence:** 3

**Summary:**

This paper proposes a framework for gait recognition using silhouette sequences and parsing sequences, XGait. This framework extracts global information through silhouette sequences. The partial information is extracted through parsing sequences, and the Part Cross-granularity Module and the Global Cross-granularity Module are designed to use the two kinds of information to complement each other. State-of-the-art results are obtained on two large datasets.

**Strengths:**

+ State-of-the-art performance is achieved on two large-scale gait datasets.
+ The first work combines silhouette sequence and parsing sequences for gait recognition, and obtains global information and local information through these two representations.
+ A neural network framework, XGait, is put forward to integrate the information of different granularity, so as to complement each other's advantages.
+ The conclusion is proved by complete experimental data and visualization results.

**Limitations:**

- The usage scenario mentioned in the title is in the wild, but Gait3D is an indoor dataset, while CCPG contains more indoor cameras than outdoor cameras, and the characteristics in the wild are not reflected in the experiment.
- An important evaluation criterion for gait recognition methods is how effective they are in cross-view. This paper has not conducted cross-view experiments on datasets such as CASIA-B.
- In Part Cross-granularity Module, how to determine the value of   is not explained clearly.

**Suitability:**

2

---

### Official Review · Reviewer_aEMH · 2024-06-04

**Rating:** 5
**Confidence:** 2

**Summary:**

The paper aims to improve gait recognition on in-the-wild datasets by combining silhouette and human parsing sequences, the two most promising gait representations.
A cross-granularity framework is proposed to exchange and fuse the complementary knowledge from these two granularity (i.e., global shape and human parts) features.
Experimental results on two large-scale in-the-wild datasets demonstrate that the proposed method achieves state-of-the-art performance, outperforming previous well-known gait recognition systems such as GaitSet, GaitBase and ParsingGait.

**Strengths:**

**Novelty**:
According to the authors, this paper is the first study to explore the complementarity of silhouette sequence and human parsing sequence for gait recognition, based on the observation that they have different granularity, information entropy and annotation quality.
The proposed GCM and PCM modules are straightforward but still somewhat novel.

**Significant results**:
The proposed method successfully achieves state-of-the-art performances on two challenging gait datasets.
Specifically, the two core designs in the paper, namely GCM and PCM, effectively improve the fusion between silhouette and parsing maps, according to Table 4 & 5.

**Limitations:**

**Additional cost**:
The paper does not clarify the computational costs (at training / inference) to run the proposed model.
According to the paper, it utilizes additional human parsing model to get parsing results, two independent GaitBase backbones during the encoding stage, and the subsequent alignment modules (GCM & PCM).

Therefore, the overall computational cost should be greater than 2 x GaitBase costs, and the comparison in Table 1 & 2 may be not fair considering this doubled overhead --- what if the baseline methods use 2x model parameters / flops?

**Some details**:
The reduction ratio $r$ for the bottleneck in the alignment MLP modules are varying on different datasets ($1$ on Gait3D, $16$ on CCPG).
As a hyper-parameter that controls network architecture (instead of some dataset-dependent one like loss weighting or image resolution), intuitively, this $r$ value should not be so sensitive to dataset choices.
The paper could be improved if the hyper-parameter searching and the analysis are presented.

**Suitability:**

2

---

### Official Review · Reviewer_T5gC · 2024-06-06

**Rating:** 4
**Confidence:** 3

**Summary:**

The paper introduces XGait, a novel gait recognition framework that integrates silhouette and parsing sequences to enhance recognition accuracy in real-world scenarios. It proposes the use of Global and Part Cross-granularity Modules to align features at different levels, demonstrating improved performance on the Gait3D and CCPG datasets.

**Strengths:**

The strengths of the paper are:

Innovation: It introduces the first gait recognition framework, XGait, that combines silhouette and parsing sequences.
Theoretical Rationale: Provides a clear theoretical basis for the fusion of features of different granularities.

Technical Implementation: Detailed description of the model architecture, including key GCM and PCM modules.
Comprehensive Evaluation: Experiments on two large-scale datasets validate the effectiveness of the approach.

Clarity: Well-structured, with figures and tables that aid understanding.

Application Potential: Implicit in discussions of robustness under challenging conditions, though not explicitly stated.

**Limitations:**

The paper's weakness is specifically in the insufficient evaluation of the model's robustness to tracking errors. There is no detailed discussion or experimental data provided on how inaccuracies in the initial frame alignment or tracking stability throughout the gait sequence affect the final recognition accuracy. Given that gait recognition in the wild often relies on video sequences with potential motion blur or occlusions, understanding the model's sensitivity to such tracking errors is essential for assessing its practical viability.

**Suitability:**

2

---

### Meta-Review · Area_Chair_vfXG · 2024-06-27

**Recommendation:** Accept (Poster)
**Confidence:** 4

**Metareview:**

All reviewers agree that this paper has somewhat novelty and vote for WA or BA. Although reviewers provided some critiques/suggestions to improve clarity in some parts, all reviewers agreed to accept it.